# Bilateral Renal Ectopia—Prenatal Diagnosis

**DOI:** 10.3390/diagnostics14050539

**Published:** 2024-03-03

**Authors:** Nicolae Gică, Livia Mihaela Apostol, Iulia Huluță, Corina Gică, Ana Maria Vayna, Anca Maria Panaitescu, Nicoleta Gana

**Affiliations:** 1Gynecology Department, Faculty of Medicine, Carol Davila University of Medicine and Pharmacy, 020021 Bucharest, Romania; gica.nicolae@umfcd.ro (N.G.); iuliahuluta16@gmail.com (I.H.); mat.corina@gmail.com (C.G.); anamariavayna@gmail.com (A.M.V.); anca.panaitescu@umfcd.ro (A.M.P.); 2Clinical Hospital of Obstetrics and Gynaecology Filantropia, 011171 Bucharest, Romania; livia-mihaela.cosma@rez.umfcd.ro

**Keywords:** renal ectopia, duplex kidney, renal anomalies

## Abstract

This report explores the diverse spectrum of congenital anomalies of the kidney and urinary tract (CAKUT), ranging from asymptomatic presentations to the most severe form characterized by bilateral renal agenesis. Genitourinary anomalies, a prevalent subset within this domain, account for a significant proportion, constituting 15–20% of anomalies identified during prenatal screening. An ectopic kidney is defined by the presence of an empty renal fossa and the displacement of the kidney from the lumbar region to alternative locations, with the pelvic region emerging as the most prevalent site. The reported case involves bilateral renal ectopia with unilateral duplex kidney. Initial suspicions of a renal anomaly arose during the first trimester, leading to a definitive diagnosis in the second trimester. The patient underwent regular monitoring every four weeks, ultimately delivering a healthy baby at term. This case underscores the frequency of renal anomalies, emphasizing that a considerable proportion remains asymptomatic. These findings contribute to a broader understanding of congenital renal anomalies, their varied manifestations, and the importance of vigilant prenatal screening for early detection and management.

Congenital anomalies of the kidney and urinary tract (CAKUT) are represented by a broad spectrum of anomalies, from asymptomatic to the most severe form, bilateral renal agenesis [1]. Genitourinary anomalies are very common, representing 15–20% of all anomalies detected prenatally [2]. The assessment of the fetal urinary tract is represented by the presence of both kidneys, their locations, the presence and shape of the bladder, and the amniotic fluid volume [2]. Ectopic kidneys refer to an empty renal fossa and the presence of the kidney in a different location other than the lumbar, with the most common location being in the pelvis. Usually, renal ectopia is unilateral or, more rarely, bilateral. Renal duplication or duplex kidney refers to an incomplete fusion of the superior and inferior poles of the moieties, resulting in complete or incomplete duplication of the collecting systems [2,3]. 

We present a case of a 35-year-old woman who came into our clinic for a first-trimester ultrasound examination at a gestational age of 12 weeks and 4 days. Her pregnancy was conceived via in vitro fertilization, using an egg donor. She had no remarkable personal or family history. The age of the egg donor was 23 years old, and she had no medical issues. The father of this baby was healthy and reported no personal or family history. First-trimester screening for chromosomal abnormalities and preeclampsia was performed using a combination of maternal age (egg donor’s age), maternal characteristics and history, ultrasound markers such as nuchal translucency, tricuspid regurgitation, ductus venosus, and nasal bone, and biochemistry (Beta-HCG and PAPP-A). The results showed a low chance of the most common trisomies (trisomies 21, 13, and 18) and a high chance of developing preeclampsia and fetal growth restriction. She started taking a prophylaxis dose of 150 mg of aspirin daily in order to reduce these risks. The patient underwent cell-free DNA testing, which showed a low chance of chromosomal abnormalities and microdeletions. The ultrasound scan at this time raised the suspicion of unilateral empty renal fossa. 

She returned at 20 weeks and 4 days for the second-trimester anomaly scan. Our careful ultrasound examination revealed bilateral empty renal fossa, a normal bladder, and normal amniotic fluid volume. No other fetal anomalies were detected, and the fetus was normally grown. Renal tissue was found in the fetal pelvis, close to the bladder and to the right (Figure 1). On a transvaginal scan, we demonstrated a duplex kidney with two renal pelvises (Figure 2a,b). The artery supply was seen coming from the internal iliac artery (Figure 3a,b). We offered invasive testing, but the patient declined. A fetal echocardiography was also performed, which showed no abnormalities of the fetal heart.

We followed the fetal growth and potential complications such as hydronephrosis or ureterocele every 4 weeks. At around 37 weeks of gestation, the patient developed polyhydramnios with an amniotic fluid index above 25, an estimated fetal weight below the 10th centile, and normal Dopplers, but no other anomalies were detected. Her glucose levels were normal throughout the pregnancy. Despite being informed that standard obstetric care usually recommends vaginal delivery in such cases, she opted for a cesarean section at 38 weeks and 3 days, resulting in the birth of a healthy 2600 g female neonate with an Apgar score of 10. The postnatal examination of the newborn confirmed the presence of a right duplex kidney located in the pelvis and a very small residual left pelvic kidney, which was not detected prenatally. 

A differential diagnosis was made with crossed fused renal ectopia or horseshoe kidney, but the presence of another small left renal kidney invalidated these diagnoses.

This case is peculiar because of the presence of both kidneys in the pelvis, the right one having an anomaly of fusion with two collecting systems and the left one being small, and the vascular supply of the right duplex kidney was aberrant, branching from the internal iliac artery.

In conclusion, renal tract anomalies are frequent, and they present in many different ways. Most of them have a good long-term prognosis. Sometimes, renal anomalies are incidental findings later on in life due to frequent urinary tract infections. Prenatal diagnosis favors adequate postnatal management.

## Figures and Tables

**Figure 1 diagnostics-14-00539-f001:**
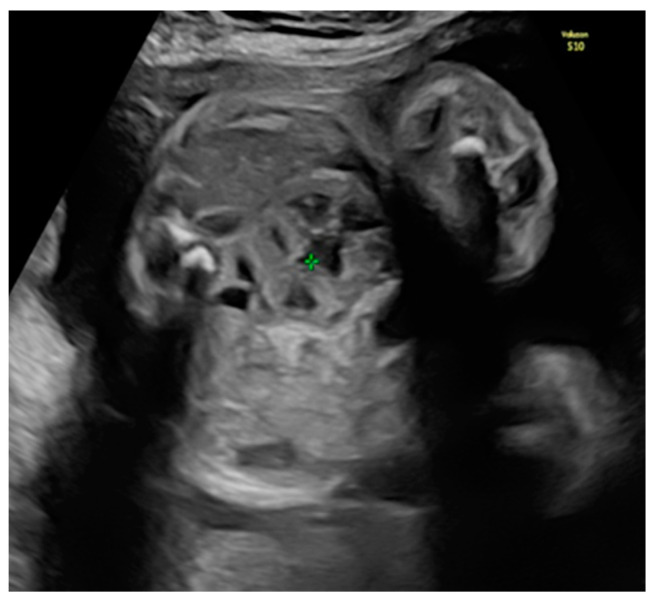
Pelvic kidney (the renal tissue is in the fetal pelvis, close to the bladder and to the right).

**Figure 2 diagnostics-14-00539-f002:**
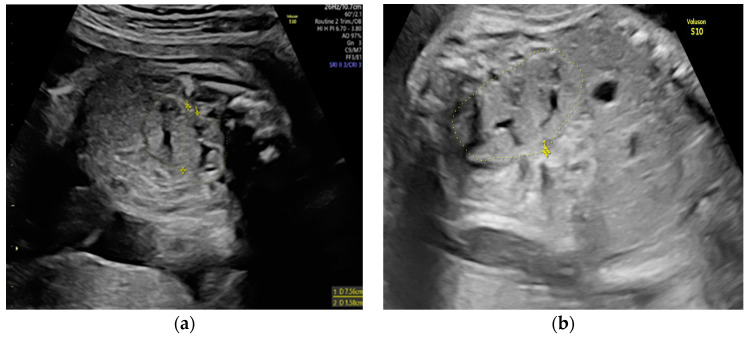
Transvaginal scan assessment: (**a**) Visualization of the right duplex kidney. (**b**) Sagittal view of the right duplex kidney.

**Figure 3 diagnostics-14-00539-f003:**
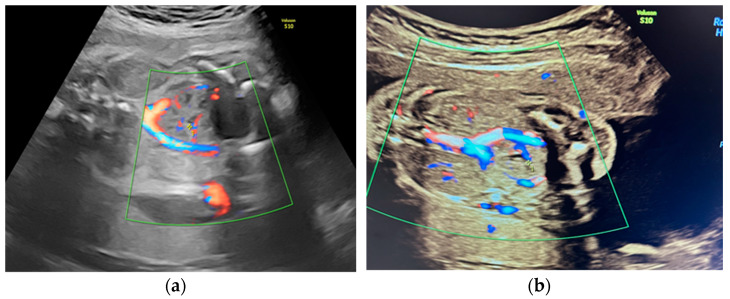
(**a**,**b**) Vascular supply of the pelvic kidney from internal iliac artery.

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
