# Peer review of "Bilateral Renal Ectopia—Prenatal Diagnosis"

_diagnostics, 2024, doi:10.3390/diagnostics14050539_

Round 1
Reviewer 1 Report
Comments and Suggestions for Authors
This case report is well-described and illustrates the ability to suspect CAKUT as soon as first trimester. Pictures are nice. The interest for the readers remains relatively low as ectopic kidneys and/or duplex kidneys are already well-known entities. I only have 2 minor comments:
-It is not clear why patient delivered at 38 GW by cesarean section (Spontaneous labour? Induction of labour? Planned CS?...)
- In the conclusion the last sentence should be replaced by "Prenatal diagnosis favours adequate postnatal management".
Author Response
I am writing to express my gratitude for the insightful comments and suggestions provided during the review process of our article titled "Bilateral Renal Ectopia - Prenatal Diagnosis." Your feedback has been invaluable in improving the quality and clarity of our work.
I am pleased to inform you that we have carefully considered all of your comments and have made the necessary revisions to the text. These changes have been highlighted for your convenience. Additionally, I would like to address the point you raised regarding the reason for the caesarean section in our case. Despite being informed that standard obstetric care usually recommends vaginal delivery in such cases, the patient opted for a caesarean section at 38 weeks and 3 days due to maternal request. It is important to note that while this was the patient's preference, it is generally recommended to follow standard obstetric care and delivery guidelines in such situations.
Once again, thank you for your valuable feedback and guidance throughout this process. We believe that the revisions made have significantly improved the manuscript, and we look forward to your continued support.
Reviewer 2 Report
Comments and Suggestions for Authors
Dr Gica et al. here describe a very interesting case of a prenatal diagnosis of bilateral renal ectopia. This type of diagnosis can be done during regular prenatal consult in pregnancy, but it can present its challenges. In this case, after the primary diagnosis and later in pregnancy, the mother presented with polyhydramnios which could have led the team to look into kidney anomalies further. The importance of this diagnosis early in life is that it helps arrange patient follow up and direct care to prevent further kidney damage, prolonging a good quality of life. This novel case helps by adding to the medical literature another verity one specialist can encounter.
Author Response
I am writing to express my gratitude for the insightful comments and suggestions provided during the review process of our article titled "Bilateral Renal Ectopia - Prenatal Diagnosis." Your feedback has been invaluable in improving the quality and clarity of our work.
Once again, thank you for your valuable feedback and guidance throughout this process. We believe that the revisions made have significantly improved the manuscript, and we look forward to your continued support.